# Ameliorative Effect of Lycopene on Follicular Reserve Depletion, Oxidative Damage, Apoptosis Rate, and Hormonal Profile during Repeated Superovulations in Mice

**DOI:** 10.3390/vetsci11090414

**Published:** 2024-09-06

**Authors:** Shimaa I. Rakha, Ahmed I. Ateya, Fatmah A. Safhi, Ahmed M. Abdellatif

**Affiliations:** 1Department of Theriogenology, Faculty of Veterinary Medicine, Mansoura University, Mansoura 35516, Egypt; shimaa_ibrahim@mans.edu.eg; 2Department of Development of Animal Wealth, Faculty of Veterinary Medicine, Mansoura University, Mansoura 35516, Egypt; 3Department of Biology, College of Science, Princess Nourah bint Abdulrahman University, P.O. Box 84428, Riyadh 11671, Saudi Arabia; faalsafhi@pnu.edu.sa; 4Department of Anatomy and Embryology, Faculty of Veterinary Medicine, Mansoura University, Mansoura 35516, Egypt

**Keywords:** lycopene, follicular reserve, oxidative stress, apoptosis, repeated superovulations

## Abstract

**Simple Summary:**

Superovulation is an important step in assisted reproductive technology that usually requires several injections of gonadotropin hormones. Repeated hormonal injections are known to badly affect the ovaries functions. This study examined the feasibility of lycopene administered in parallel with repeated hormonal injections in rescuing the ovarian damage caused by four superovulatory cycles. To test this hypothesis, we compared ovaries and blood samples collected from mice injected four times with lycopene (R4-Lyc) or four times with saline (R4). Our analysis included the number of ovarian follicles, the expression of genes related to ovarian antioxidation and cell death, and the serum levels of the female reproductive hormones. Our results showed that the ovaries in the R4-Lyc group contained fewer unhealthy ovarian follicles and more abundant healthy ones than those in the R4 group. Additionally, lycopene-treated mice ovaries showed increased total antioxidant capacity, decreased H_2_O_2_ concentrations, greater levels of antioxidant genes, and reduced levels of the cell death-associated genes. In terms of reproductive hormones, lycopene injections improved serum levels of progesterone, estradiol, and inhibin-B. According to the current study, lycopene has a major role in reducing ovarian damage caused by repeated hormonal injections and may also aid in improving the success of in vitro embryo production.

**Abstract:**

Superovulation is a crucial step in assisted reproductive technology that involves the administration of gonadotrophins. Repeated superovulations result in severe ovarian damage. The present study investigated the effect of in vivo administration of lycopene on ovarian damage induced by four successive cycles of superovulation. Superovulated mice were simultaneously administered intraperitoneally with saline (R4) or 5 mg/kg lycopene (R4-Lyc). The evaluated parameters were the count of different types of follicles, expression of ovarian antioxidant- and apoptosis-related genes, and serum concentrations of estradiol, progesterone, and inhibin-B. Increased numbers of healthy follicles and a decreased count of atretic follicles were observed in mice of the R4-Lyc group compared to those of the R4 group. Moreover, significantly higher mRNA levels of *Sod3*, *Cat*, and *Nrf2* and lower mRNA levels of *Keap1*, *Tnf*, *Nfkb*, and *Casp3*, together with decreased H_2_O_2_ concentrations and increased total antioxidant capacity, were detected in the ovaries of lycopene-treated mice. Regarding serum reproductive hormones, elevated concentrations of estradiol, progesterone, and inhibin-B were evident in lycopene-administered mice. The present study reports a significant role of lycopene in alleviating the ovarian damage induced by multiple hormonal superstimulations, which could help to improve the outcomes of in vitro embryo production.

## 1. Introduction

Superovulation is a crucial step in assisted reproductive technology. It is usually achieved via the administration of exogenous gonadotrophins. The administered gonadotrophins induce a progressive inflammatory reaction in the growing ovarian follicles that eventually results in the rupture of the mature follicles at the surface of the ovary and the release of the ova into the lumen of the oviduct [1]. Previous research has shown that repeated superovulations have harmful effects on the ovaries. Multiple rounds of superovulation aggravate the ovarian damage induced by gonadotrophins through excessive and sustained production of inflammatory mediators, including prostaglandins, histamine, and bradykinin [2]. Moreover, the cumulative damage induced by repeated exposure to gonadotropins has been reported to significantly contribute to ovarian aging through depletion of the follicle stores [3,4].

The balance in the redox status is important for the survival and homeostasis of the living cells. Such balance is usually achieved by a number of antioxidant enzymes that act continuously to eliminate excess levels of cell oxidation products and preserve mitochondrial health. The process of ovulation elicits higher rates of production of oxidative stress products, including superoxide radicals, hydrogen peroxides, lipid peroxides, and others [5]. Repeated ovarian stimulations using exogenous hormones are, thus, in turn, associated with increased buildup of oxidative damage products and mitochondrial DNA mutants in ovaries [6]. Sustained production of oxidative damage products in the female reproductive tract adversely affects the developmental competence of the produced oocytes and embryos [7], as well as the follicular count, via increasing the rates of follicular atresia [8].

Mice subjected to repeated cycles of superovulation (≥four times) revealed higher levels of oxidative stress, abnormal spindle formations, and mitochondrial damage in oocytes and their surrounding cumulus cells compared to their non-superovulated counterparts [6,9,10]. The use of antioxidants during the induction of superovulation represents a promising trend to alleviate the hormone-induced damage to the ovaries and oocytes. For instance, simultaneous administration of melatonin during superovulation in mice resulted in a significant decrease in mitochondrial and DNA damage and prevented apoptosis of oocytes [11]. Furthermore, a restorative effect of *Lycium barbarum* polysaccharide on mouse ovarian injuries induced by multiple superovulations, in terms of follicular count and oocyte quality, has been shown [12]. The latter effect was also associated with declined levels of ovarian oxygen free radicals and lipid peroxides.

Lycopene is a red-colored carotenoid pigment found in tomatoes and a number of fruits and vegetables [13]. Lycopene has a powerful free radical scavenging ability, mainly attributed to its high content of conjugated double bonds [14,15]. Our previous studies have shown that lycopene delays the aging of mouse oocytes and enhances their in vitro maturation rates, which are associated with improvements in the oocytes’ oxidative status [16,17]. According to a recent survey, lycopene intake is associated with a lower risk of female infertility in humans [18]. Moreover, its administration during the course of chemotherapeutic treatment protected against the ovarian damage induced by cisplatin in rats by activating the ovarian antioxidant mechanisms in rats [19]. Given this and in search of factors that could mitigate the damage induced by repeated superovulation, the present study aimed to investigate the effect of in vivo administration of lycopene on the ovarian damage induced by four successive cycles of superovulation. Mice with single ovarian hyperstimulation, known to have minimal ovarian damage, were used as controls to scale the effects gained by lycopene supplementation. The evaluated parameters included the count of different types of follicles, the expression of the ovarian antioxidant- and apoptosis-related genes, as well as the serum concentrations of important female reproductive hormones.

## 2. Materials and Methods

### 2.1. Chemicals

Lycopene extract (≥98%; M.W. = 536.87 g/mol) was purchased from Nawah Scientific Inc. (HIKA2010, Cairo, Egypt). A stock solution of 1 mM lycopene was prepared by dissolving lycopene powder in dimethyl sulfoxide (DMSO, D8418, Sigma-Aldrich, St. Louis, MO, USA) as previously described [16,17]. Lycopene stock solution was immediately aliquoted, protected from light, and stored at −20 °C until the day of injection. Pregnant mare serum gonadotrophin (PMSG; Gonaser, HIPRA, Amer, Spain) and human chorionic gonadotropin (hCG, Nile for Pharmaceuticals & Chemical Industries, Cairo, Egypt) were used to induce superovulation in mice.

### 2.2. Animals

The present study was permitted by the Committee for Research Ethics at the Faculty of Veterinary Medicine, Mansoura University, Egypt (Code No: VM.R.24.01.148). The study procedures complied with the ARRIVE guidelines and were performed according to the National Institutes of Health guide for the use of animals in research (NIH Publications No. 8023, revised 1978). Sixty adult Swiss albino CD1 strain mice aged 8–10 weeks old (25 ± 3.25 g) were used. The mice were obtained from the animal unit of Mansoura University, Egypt. The animals were housed in separate cages (five mice per cage), maintained under standard laboratory conditions in a temperature-controlled environment (21–23 °C) under a 12 h light/dark cycle, and gained ad libitum access to water and commercially available rodent pellet diets (Meladco, El-Obour City, Cairo, Egypt). To evaluate the effect of lycopene on the ovarian damage induced by repeated superovulation in mice, the animals were allocated into three experimental groups, as shown in Section 2.3.

### 2.3. Experimental Design

After 2 weeks of acclimatization, three experimental groups (20 each; four cages per group) were set as follows: R1 (superovulated once, PMSG (10 IU, i.p.) at day 1 and hCG (10 IU, i.p.) at day 3); R4 (superovulated four successive times); and R4-Lyc (superovulated four successive times and received lycopene i.p. at a dose of 5 mg/kg body weight simultaneously). The lycopene dose was based on previous reports about lycopene’s effective daily requirements [20,21]. All animals received injections on a weekly basis for four consecutive weeks. As high plasma levels of lycopene were proven to be maintained for up to seven days following its administration [22], the timing used for lycopene injections in the present study ensured continuous exposure of the ovary to it. To avoid the influence of stress induced by injection, mice of the R1 group were injected with saline twice weekly for the first three weeks. The experimental groups of the present study are summarized in Table 1.

### 2.4. Animals’ Euthanasia, Blood, and Tissue Sampling

The animals of all groups were euthanized by a skilled researcher (AA) using the cervical dislocation method 26 h following the last hCG injection. Blood samples were withdrawn from all mice through the retro-orbital sinus just before euthanasia. To minimize mice suffering during blood collection, a topical eye anesthetic solution (tetracaine 0.5%) was applied prior to the blood collection. The abdominal cavity of each mouse was opened, and the ovaries were extracted and weighed. For each experimental group, 12 ovaries (four per cage; three different cages) were preserved in 10% neutral buffered formalin and subjected to histological and immunohistochemical analysis; 12 ovaries were used for biochemical assessment of the redox status of the ovaries, and the remaining 16 ovaries were kept at −80 °C for gene expression analysis.

### 2.5. Histological and Immunohistochemical Analyses

Formalin-fixed mouse ovaries were dehydrated in ascending grades of ethanol and finally embedded in molten paraffin [23]. The paraffin blocks were serially and longitudinally cut into 5 µm thick sections spanning the whole ovary; one of every 15 sections was collected and stained with Harris hematoxylin and eosin stain. The total number of healthy (primordial, primary, secondary, and antral) and atretic follicles was quantified and compared between the three groups. Cleaved caspase 3 immunohistochemistry was performed on five sections cut 100 µm apart per ovary, as previously described [24,25]. Caspase 3 immunoreactive follicles were counted in all immunostained sections, regardless of the number of follicular cells expressing caspase 3.

### 2.6. Biochemical Assessment of the Ovarian Redox Status

The ovaries were lysed in lysis buffer (50 mM sodium phosphate, 300 mM NaCl, pH = 8.0), centrifuged at 12,000 rpm for 15 min at 4 °C, and the supernatant was collected and used. Measurement of H_2_O_2_ levels and total antioxidant capacity (TAC) was achieved using commercially available kits (Biodiagnostics, Cairo, Egypt) per the manufacturer’s instructions.

### 2.7. Gene Expression Analysis

Trizol reagent was utilized to isolate the total RNA from ovarian tissue specimens according to the manufacturer’s instructions (Direct-zolTM RNA MiniPrep, catalog No. R2050, Zymo Research, Tustin, CA, USA). The quantity and purity of RNA were measured using a Nanodrop spectrophotometer (UV-Vis spectrophotometer Q5000, Quawell Technology, San Jose, CA, USA), and its integrity was evaluated by gel electrophoresis. The RNA samples were then reverse-transcribed into cDNA per the manufacturer’s protocol (SensiFastTM cDNA synthesis kit, catalog No. Bio-65053, Bioline Ltd., London, UK). The reaction mixture was performed in a total volume of 20 µL consisting of total RNA up to 1 µg, 4 µL 5× Trans Amp buffer, 1 µL reverse transcriptase, and DNase-free water up to 20 µL. The final reaction mixture was placed in a thermal cycler, and the following program was used: primer annealing at 25 °C for 10 min, reverse transcription at 42 °C for 15 min, followed by inactivation at 85 °C for 5 min. The samples were kept at 4 °C.

The relative mRNA abundance of the ovarian antioxidant (*Sod3*, *Cat*, *Nrf2*, and *Keap1*) and inflammatory (*Tnf*, *Nfkb*, and *Casp3*) factors were determined by quantitative real-time PCR (qRT-PCR) using SYBR Green PCR Master Mix (2× SensiFastTM SYBR, catalog No. Bio-98002, Bioline Ltd., London, UK). The housekeeping gene β-actin was used as an internal control. The details of the primer used for qRT-PCR analysis are shown in Table 2. The reaction mixture was carried out in a total volume of 20 µL containing 10 µL 2× SensiFast SYBR, 3 µL cDNA, 5.4 µL H_2_O (d.d. water), and 0.8 µL of each primer. Real-time PCR conditions were performed as follows: 95 °C for 4 min followed by 40 cycles of 94 °C for 15 s; annealing temperatures are shown in Table 2 for 30 s, and extension temperature at 72 °C for 20 s. At the end of the amplification phase, a melting curve analysis was performed to verify the specificity of the PCR products. To normalize target gene expression levels, the relative expression of each gene in every sample versus the control in comparison to the *β-actin* gene was calculated according to the 2^−ΔΔCt^ method [26].

### 2.8. Serum Hormone Levels Analysis

Blood samples were centrifuged at 4000 rpm for 10 min, and the serum was decanted. Levels of estradiol, progesterone (Cayman Chemicals, Ann Arbor, MI, USA), and inhibin B (Cusabio, Wuhan, China) were estimated using commercial ELISA kits according to the manufacturer’s instructions. The detection range and sensitivity of the used assays were 0.61–10,000 pg/mL and 20 pg/mL; 7.8–1000 pg/mL and 10 pg/mL; 1–200 pg/mL and 0.5 pg/mL for estradiol, progesterone, and INH-B, respectively.

### 2.9. Statistical Analysis

Data were analyzed using the GraphPad Prism 7 (GraphPad Software, La Jolla, CA, USA) and expressed as means ± SEM. Differences between groups were determined using one-way ANOVA followed by Tukey’s multiple comparison test. *p* < 0.05 was used to express statistical significance.

## 3. Results

Overall, increased numbers of corpora lutea were noted in the ovaries of mice that received four successive superovulations with (R4-Lyc) or without lycopene (R4) compared to those that received a single superovulation (R1) (Figure 1A,D,G). Moreover, mice of the R4-Lyc group displayed better ovarian morphology than those of the R4 group, especially in terms of follicular appearance (Figure 1B,C,E,F,H,I).

A quantitative comparison of the body weight of mice did not reveal significant differences between the three experimental groups (Figure 2A). On the other hand, the ovarian weight was higher in the ovaries of mice that received four successive superovulations compared to those that received a single superovulation (Figure 2B). Notably, increased total numbers of primordial, primary (Figure 2C), secondary (Figure 2D), and antral follicles (Figure 2E), along with a decreased total number of atretic follicles (Figure 2F), were evident in mice of the R4-Lyc group compared to those of the R4 group.

To study the effect of lycopene supplementation on the ovarian redox status of mice subjected to repeated superovulations, the gene expression levels of oxidative stress-related genes were examined. The ovarian H_2_O_2_ concentration and total antioxidant capacity were also estimated. A significant rise in the mRNA expression levels of the antioxidant enzymes superoxide dismutase 3 (*Sod3*), catalase (*cat*), and nuclear factor erythroid 2-related factor 2 (*Nrf2*), together with a significant decline in Kelch-like ECH-associated protein 1 (*Keap1*) mRNA levels, were seen in the ovaries of the R4-Lyc group compared to those of the R4 group (Figure 3A–D). These changes in gene expression levels were also coupled with a decreased ovarian H_2_O_2_ concentration and an increased ovarian total antioxidant capacity (Figure 3E,F).

To evaluate the effect of lycopene supplementation on the apoptotic damage induced by repeated superovulations in mice ovaries, the expression levels of apoptosis-related genes tumor necrosis factor (*Tnf*), nuclear factor-κB (*Nfkb*), and caspase 3 (*Casp3*) were examined. A quantitative comparison of caspase 3 immunoreactive (IR) follicles was also performed. A significant decrease in the mRNA expression levels of *Tnf*, *Nfkb*, and *Casp3* was evident in the ovaries of the R4-Lyc group compared to those of the R4 group (Figure 4A–C). These changes in gene expression levels were paralleled with a decline in the number of caspase 3 IR follicles (Figure 4D). Fractionation of caspase 3 IR follicles showed that the apoptotic damage mainly involved the follicles of the secondary order and appeared to be concentrated within the granulosa cells of the follicle (Figure 4E–H).

Regarding the effect of lycopene supplementation on the levels of female reproductive hormones during repeated superovulations, higher serum concentrations of estradiol, progesterone, and inhibin B were detected in mice of the R4-Lyc group compared to those of the R4 group (Figure 5A–C). The intra- and inter-assay coefficients of variability (%) of the used tests were 8.2 and 11 for estradiol, 7.5 and 12.6 for progesterone, and 7.9 and 14 for INH-B.

## 4. Discussion

Superovulation is a routine step during assisted reproductive technology in humans and animals in which gonadotrophic hormones are used to stimulate the ovaries to produce more oocytes. Depending on the degree of infertility and several other related circumstances, multiple cycles of superovulation are usually employed. However, repeated successive cycles of superovulation are known to result in severe ovarian damage due to increased mitochondrial damage, excessive production of ROS, exhaustion of ovarian antioxidant mechanisms, and depletion of ovarian follicle reserves [10,27,28]. In the present study, we investigated the deleterious effects of repeated superovulation on mouse ovaries and the possible protective role of lycopene in ameliorating such effects. The follicular count, especially those of secondary order, was significantly decreased. Moreover, declined levels of antioxidant factors and elevated levels of ROS, inflammatory cytokines, and apoptosis-associated factors were dominant features in the ovaries of superovulated mice. Lycopene simultaneous supplementation during repeated cycles of ovarian hyperstimulation mitigated the damage induced by exogenous hormones on the ovaries, along with an improvement in the reproductive hormonal status of the mice.

Ovarian damage associated with repeated superovulation persists for a long time. For instance, Rhesus monkeys subjected to four cycles of hormone-based stimulations displayed mitochondrial ultrastructural abnormalities involving the granulosa cells of the ovarian follicles, together with reduced expression of genes responsible for steroid hormone biosynthesis, including *STAR* (steroidogenic acute regulatory protein) and *CYP19A1* (cytochrome P450 family 19 subfamily A member 1), for five years following the ovarian stimulation [29]. Moreover, repeated superovulation has been reported to induce ovarian aging. Among the factors with increased expression are the senescence-associated factors *P16* or *Cdkn2a* (cyclin-dependent kinase inhibitor 2A) [30] and *P53* or *Trp53* (transformation-related protein 53) [31].

The use of antioxidant substances to protect against ovarian damage induced by repeated cycles of ovarian hyperstimulation is under active investigation by several researchers. Epicatechin is a polyphenolic compound present mainly in woody plants. Intraperitoneal injection of epicatechin during repeated superovulation (up to five cycles) in mice increased the gene expression of antioxidant enzymes (*Gpx1*, *Prdx3*, and *Sod*) and enhanced the total antioxidant capacity of the ovaries [32]. He’s Yangchao Recipe, a mixture of eight Chinese herbs, has been found to lessen the adverse effects of four cycles of repeated superovulation in aged mice by preserving the number of growing follicles and decreasing the rate of granulosa cell turnover [31]. In line with these studies, a promising role for lycopene in mitigating the deleterious effects of repeated superovulation on the murine ovaries has been suggested by the present work. Lycopene maintained a significantly higher number of ovarian follicles and achieved higher gene expression of antioxidant enzymes (*Sod3*, *Cat*, *Nrf2*, and *Keap1*) in the ovaries of lycopene-treated mice than those of non-treated controls.

Increased granulosa cell apoptosis is commonly encountered following repeated cycles of superovulation [30]. Apoptosis involves the activation of several proteases, including caspases, following the release of specific proapoptotic cytokines. Among those cytokines is tumor necrosis factor (Tnf). Increased expression of Tnf has been reported to be associated with arrested development of the ovarian follicles and decreased production of the female reproductive hormone from the murine as well as the bovine ovaries [33,34]. Inhibition of Tnf activity in granulosa cells has been reported to improve their steroidogenic activity [33]. Thus, targeting Tnf in granulosa cells could represent a novel strategy to improve the hormonal secretory activity of granulosa cells. In this regard, lycopene has been shown to regulate the expression of Tnf in both reproductive [35,36] and non-reproductive tissues [37]. In view of this, the downregulated ovarian expression of Tnf observed in the present study upon lycopene treatment could account for the higher count of ovarian follicles that survived the repeated damage induced by exogenous gonadotropins. 

Granulosa cells of the ovarian follicles play an indispensable role in the production of female sex hormones. Therefore, maintaining a sufficient number of healthy follicles is fundamental to achieving normal levels of female hormones. An inhibitory effect of lycopene on the rate of follicular apoptosis, as shown by the decreased expression of the apoptosis-related gene *Casp3*, has been detected in this study. To investigate the distribution of apoptotic cells across different types of follicles, Casp3 immunohistochemistry was employed. Our immunohistochemical analysis showed the differences in Casp3 immunoreaction in granulosa cells were mainly affecting those of the secondary order, with modest expression in those of the primordial and primary orders. In line with this observation, Fenwick and Hurst (2002) [38] reported Casp3 to be absent in ~90% of primordial and primary follicles, and its expression in higher-order follicles could indicate the initiation of their atresia. Since most of the sex hormones are produced by ovarian follicles of secondary and mature follicles, apoptotic damage involving the granulosa cells of these follicles during repeated cycles of superovulation may account for the decline in sex hormone production by them.

The serum reproductive parameters are widely used to monitor the function of the ovaries. Serum concentrations of the sex hormones estradiol, progesterone, and inhibin-B are classically used to evaluate ovarian activity [39]. Earlier reports have shown consecutive superovulations to inhibit the production of estradiol in granulosa cells [30,40]. The diminished levels of these hormones in superovulated animals are primarily driven by the accumulation of an excessive amount of ROS generated during ovarian superstimulation, leading to accelerated apoptosis of granulosa cells [41]. This study revealed that repeated superovulations affect the function of the ovaries in female mice by decreasing estradiol, progesterone, and inhibin-B. These findings suggest that consecutive superovulations may affect ovarian endocrine function. Augmented levels of estradiol, progesterone, and inhibin-B were detected in the lycopene-treated group compared to controls. The positive effect of lycopene on reproductive hormone status in superovulated mice could be a result of its potent ROS-scavenging activity [42].

## 5. Conclusions

The present study reported a significant role for lycopene in alleviating the ovarian damage induced by repeated cycles of hormonal superstimulation. A protective effect of lycopene on the count of ovarian follicles and the production of reproductive hormones by the ovaries has been shown to be associated with enhanced ovarian antioxidant capacity and reduced rates of follicle apoptosis. Further studies are needed to decipher the mechanisms mediating lycopene effects during hormone stimulation and to test the possible synergistic roles of other natural antioxidants.

## Figures and Tables

**Figure 1 vetsci-11-00414-f001:**
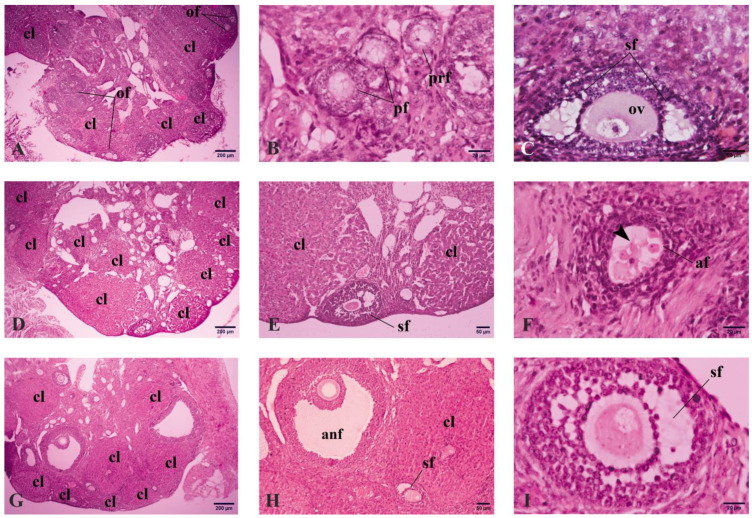
Representative photomicrographs for the ovarian microanatomy in mice that received a single superovulation (R1; (**A**–**C**)) or four superovulations without (R4; (**D**–**F**)) or with lycopene treatment (R4-Lyc, (**G**–**I**)). Abbreviations: af, atretic follicle; cl, corpus luteum; of, ovarian follicles; ov, ovum; pf, primary follicles; prf, primordial follicles; sf, secondary follicles; anf, antral follicles. The arrowhead in F denotes the wrinkled zona pellucida of an atretic follicle.

**Figure 2 vetsci-11-00414-f002:**
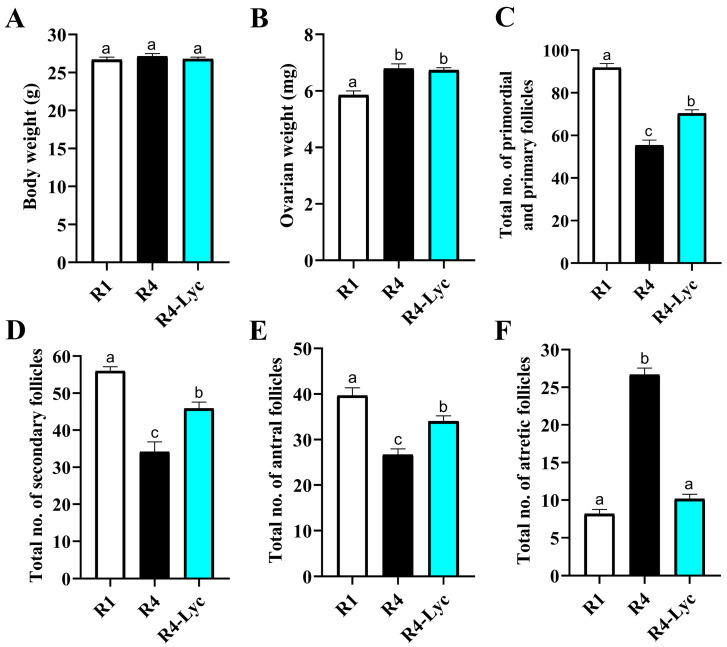
Changes in body weight (**A**), ovarian weight (**B**), total number of primordial and primary follicles (**C**), total number of secondary follicles (**D**), total number of antral follicles (**E**), and total number of atretic follicles (**F**) of mice that received a single superovulation (R1) or four superovulations without (R4) or with lycopene supplementation (R4-Lyc). Data are presented as mean ± SEM. Different superscript letters indicate statistical significance (*p* < 0.05).

**Figure 3 vetsci-11-00414-f003:**
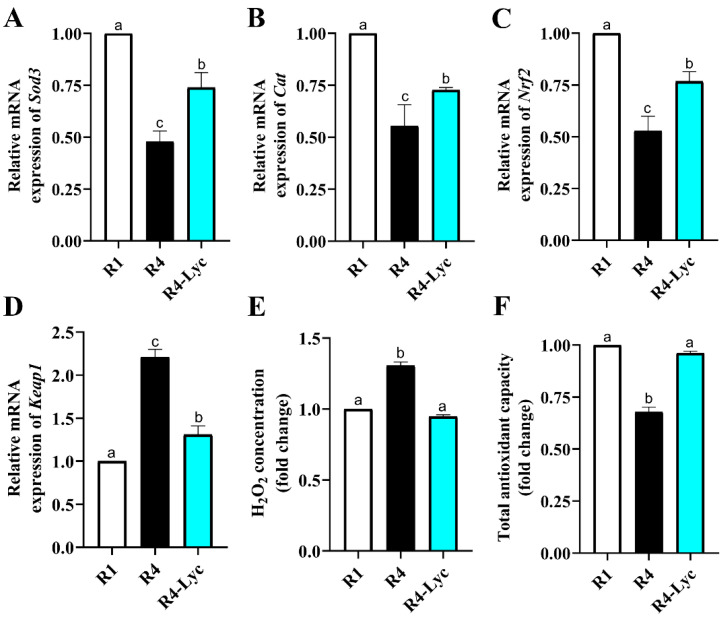
Effect of lycopene supplementation on the ovarian redox status of mice that received a single superovulation (R1) or four superovulations without (R4) or with lycopene supplementation (R4-Lyc). (**A**) Changes in mRNA expression of superoxide dismutase 3 (*Sod3*). (**B**) Changes in mRNA expression of catalase (*cat*). (**C**) Changes in mRNA expression of nuclear factor erythroid 2-related factor 2 (*Nrf2*). (**D**) Changes in mRNA expression of Kelch-like ECH-associated protein 1 (*Keap1*). (**E**) Changes in ovarian H_2_O_2_ concentration. (**F**) Changes in ovarian total antioxidant capacity. Data are presented as mean ± SEM. Different superscript letters indicate statistical significance (*p* < 0.05).

**Figure 4 vetsci-11-00414-f004:**
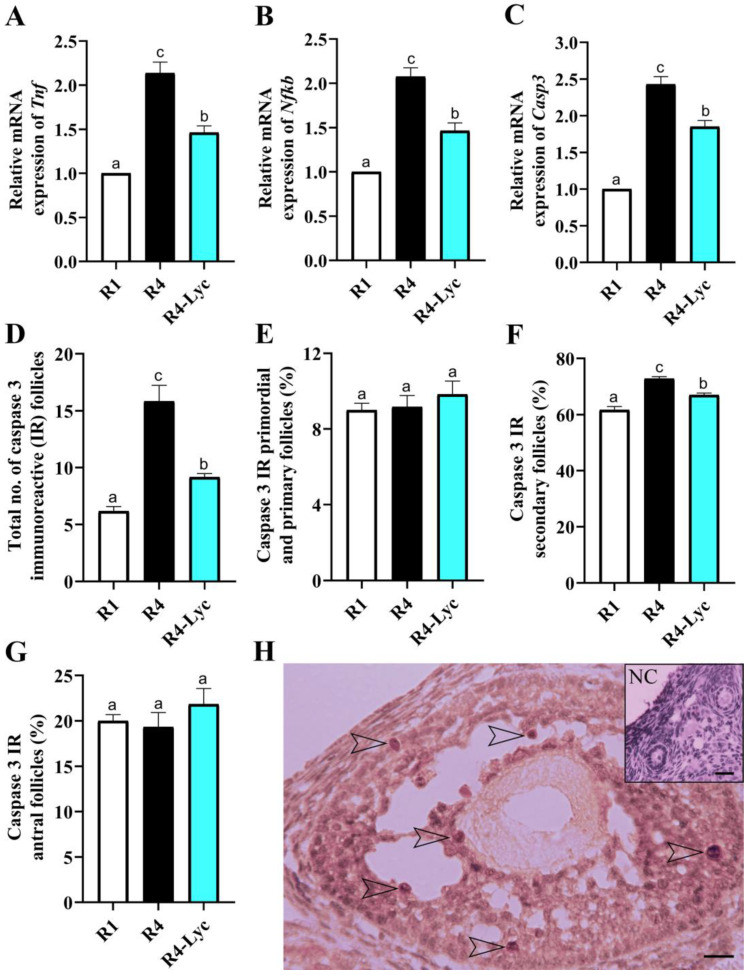
Effect of lycopene supplementation on the ovarian apoptotic damage of mice that received a single superovulation (R1) or four superovulations without (R4) or with lycopene supplementation (R4-Lyc). (**A**) Changes in mRNA expression of tumor necrosis factor (*Tnf*). (**B**) Changes in mRNA expression of nuclear factor-κB (*Nfkb*). (**C**) Changes in mRNA expression of caspase 3 (*Casp3*). (**D**) Changes in total number of caspase 3 immunoreactive (IR) follicles. (**E**) Changes in percentages of caspase 3 IR primordial and primary follicles. (**F**) Changes in percentages of caspase 3 IR secondary follicles. (**G**) Changes in percentages of caspase 3 IR antral follicles. (**H**) Caspase 3 immunoreaction in a secondary follicle; the reaction is mainly localized to the granulosa cells of the follicle (empty arrowheads). NC, negative control. Scale bars = 20 µm. Data are presented as mean ± SEM. Different superscript letters indicate statistical significance (*p* < 0.05).

**Figure 5 vetsci-11-00414-f005:**
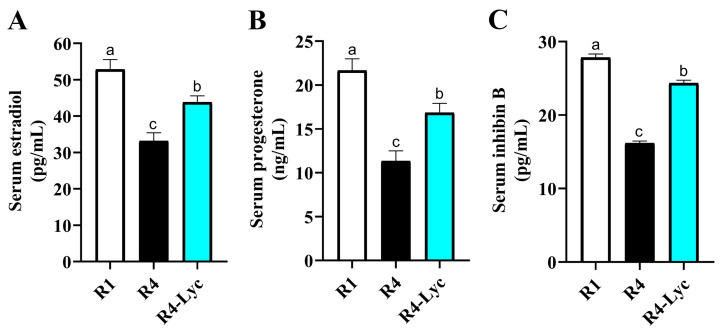
Effect of lycopene supplementation on the serum levels of female reproductive hormones in mice that received a single superovulation (R1) or four superovulations without (R4) or with lycopene supplementation (R4-Lyc). (**A**) Changes in serum estradiol levels. (**B**) Changes in serum progesterone levels. (**C**) Changes in serum inhibin B levels. Data are presented as mean ± SEM. Different superscript letters indicate statistical significance (*p* < 0.05).

**Table 1 vetsci-11-00414-t001:** Summary of experimental groups.

Group	Week 1		Week 2		Week 3		Week 4	
R1	saline	saline	saline	saline	saline	saline	PMSG	hCG
R4	PMSG	hCG	PMSG	hCG	PMSG	hCG	PMSG	hCG
R4-Lyc	PMSG; lycopene	hCG; lycopene	PMSG; lycopene	hCG; lycopene	PMSG; lycopene	hCG; lycopene	PMSG; lycopene	hCG; lycopene

R1 group: mice were superovulated once, PMSG (10 IU, i.p.) at day 1 and hCG (10 IU, i.p.) at day 3. R4 group: mice were superovulated four successive times. R4-Lyc group: mice were superovulated four successive times and received lycopene i.p. at a dose of 5 mg/kg body weight simultaneously. PMSG, pregnant mare serum gonadotrophin. hCG, human chorionic gonadotropin. Lyc, lycopene.

**Table 2 vetsci-11-00414-t002:** Oligonucleotide-based real-time PCR primers for the investigated antioxidant and inflammatory genes.

Researched Marker	Primer	Product Size (bp)	Annealing Temp. (°C)	GenBank Isolate
*Sod3*	F5′-TTCTACGGCTTGCTACTGGC-3′ R5′-GCTAGGTCGAAGCTGGACTC-3′	74	60	NM_011435.3
*Cat*	F5′-CACTGACGAGATGGCACACT-3′ R5′-TGTGGAGAATCGAACGGCAA-3′	175	60	NM_009804.2
*Nrf2*	F5′-CCTCACCTCTGCTGCAAGTA-3′ R5′-TCAAATCCATGTCCTGCTGGG-3′	120	58	NM_010902.5
*Keap1*	F5′-TCGTAGGGTGGTGGCCG-3′ R5′-ATGGGGTTCCGGATGACAAG-3′	78	60	AB020063.1
*Tnf*	F5′-GCCTCTTCTCATTCCTGCTTGT-3′ R5′-CACTTGGTGGTTTGCTACGACG-3′	203	58	AY423855.1
*Nfkb*	F5′-CTCTGGCACAGAAGTTGGGT-3′ R5′-CCCGGAGTTCATCTCATAGTTGT-3′	101	58	NM_001410442.1
*Casp3*	F5′-GGGGAGCTTGGAACGCTAAG-3′ R5′-CCGTACCAGAGCGAGATGAC-3′	232	58	NM_009810.3
*β-actin*	F5′-AGGGAAATCGTGCGTGACAT-3′ R5′-TCCAGGGAGGAAGAGGATGC-3′	59	60	AY618569.1

F, forward; R, reverse; *Sod3*, superoxide dismutase 3; *Cat*, catalase; *Nrf2*, nuclear factor erythroid 2-related factor 2; *Keap1*, Kelch-like ECH-associated protein 1; *Tnf*, tumor necrosis factor alpha; *Nfkb*, nuclear factor kappa B; *Casp3*, caspase 3; *β-actin*, beta-actin.

## Data Availability

All data are available from the corresponding author on request.

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
