# Peer review of "Ameliorative Effect of Lycopene on Follicular Reserve Depletion, Oxidative Damage, Apoptosis Rate, and Hormonal Profile during Repeated Superovulations in Mice"

_vetsci, 2024, doi:10.3390/vetsci11090414_

Round 1
Reviewer 1 Report
Comments and Suggestions for Authors
This manuscript is well-written and the result is intersting. However, some minor revisions are needed.
1. Please provide the main indicators of methodological valuation for serum hormones levels analysis, for example, sensitivity, intro-assay and inter-assay CV.
2. Please put scale bar into the figure 4H, and also the negative control of immunohistochemistry.
Author Response
Comment: This manuscript is well-written, and the result is interesting. However, some minor revisions are needed.
Response: We would like to thank our reviewer for the insightful comments that will definitely help to improve the quality of our work. We hereby provide a detailed response to your concerns.
Comment: Please provide the main indicators of methodological valuation for serum hormones levels analysis, for example, sensitivity, intro-assay and inter-assay CV.
Response: Per our reviewer’s suggestion, details on methodological valuation of serum reproductive hormone levels were added to the manuscript. The range of detection and sensitivity of the used kits were added to the methodological section (please see lines 225-227 of the revised version of the manuscript). Moreover, the intra- and inter-assay CVs (%) of the used tests were included in the relevant part of the results section (please see lines 300-302 of the revised version of the manuscript).
Comment: Please put scale bar into the figure 4H, and also the negative control of immunohistochemistry.
Response: Thank you for your notice. A scale bar has been added to Figure 4H. A picture of the negative control section was also added to Figure 4 (please see line 295 of the revised version of the manuscript).
Reviewer 2 Report
Comments and Suggestions for Authors
I have only a comment for the authors:
Lines 23-24, 39-40: Lycopene‐treated miceʹs ovaries showed decreased total antioxidant capacity, increased H2O2 concentrations. However, the graphs in the figure 3 (E) Changes in ovarian H2O2 concentration. F) Changes in ovarian total antioxidant capacity). The results showed that single superovulation was similar to lycopene treatment.
Author Response
Comment: I have only a comment for the authors.
Response: We would like to thank our reviewer for careful reading of our work. We hereby provide a detailed response to your concerns.
Comment: Lines 23-24, 39-40: Lycopene‐treated miceʹs ovaries showed decreased total antioxidant capacity, increased H2O2 concentrations. However, the graphs in the figure 3 (E) Changes in ovarian H2O2 concentration. F) Changes in ovarian total antioxidant capacity). The results showed that single superovulation was similar to lycopene treatment.
Response: Thank you very much for pointing out these typing errors. We have corrected the text of the indicated parts to match our findings as follows:
Lines 24-25: Additionally, lycopene-treated mice's ovaries showed increased total antioxidant capacity, decreased H2O2 concentrations …
Lines 39-41: … together with decreased H2O2 concentrations and increased total antioxidant capacity, were detected in the ovaries of lycopene-treated mice.
Reviewer 3 Report
Comments and Suggestions for Authors
The paper is well written. I mainly have some comments that may help the authors to add some information that is currently missing.
1. The simple summary is not really simple and has a large overlap with the abstract. Try to rewrite the summary in simple terms.
2. In the M&M, the authors refer to the ARRIVE Guidelines, which is applaudable. However, I do miss some aspects in the M&M that are essential when writing according to the ARRIVE guidelines. Below, I list a few that I noticed. I advise the authors to take the ARRIVE guidelines at hand and check for any other information that may be missing:
- Full (sub)strain name of the Swiss mice, and where they were purchased or bred.
- Type of diet the mice were fed
- Experimental set-up: 60 mice were used, however, the experimental design only accounts for 3 groups of 15 mice each. What happened to the other 15 mice?
- Information on randomisation is missing. This is especially important for the randomisation of ovaries that are attributed to 3 groups, since each mouse has only 2 ovaries. It is also important for the 5 mice in each cage: were they randomly assigned to the different treatment groups, or were whole cages assigned to one group? This may affect the experimental unit. If whole cages were assigned to an experimental group, but still mice were seen as an experimental unit, some information is necessary describing how the authors interpret the risk of pseudo replication for the parameters they measure.
- Information on blinding is missing: Were people injecting the mice blind for treatment? Were people analysing the ovaries blind for treatment?
- Information on acclimitisation is missing
- Statistics: for all comparisons, one-way anova with Tukeys post-hoc test were used. Did all parameters meet the assumptions for one way anova? Eg with regard to normal distribution? If not, what was done? Please provide some more info for each of the parameters measured.
3. Ethical remark: Prior to euthanasia, mice were bled through retro-orbital puncture. According to good practice, this method is only acceptable under terminal anesthesia. Was terminal anesthesia applied? If not, why not?
Author Response
Comment: The paper is well written. I mainly have some comments that may help the authors to add some information that is currently missing.
Response: We would like to thank our reviewer for the insightful comments that will definitely help to improve the quality of our work. We hereby provide a detailed response to your concerns.
Comment: The simple summary is not really simple and has a large overlap with the abstract. Try to rewrite the summary in simple terms.
Response: Thank you for your comments. We have revised the simple summary, trimmed several parts of it, and replaced certain scientific terms with regular ones. These modifications are believed to make the summary easier to understand by general readers.
Comment: In the M&M, the authors refer to the ARRIVE Guidelines, which is applaudable. However, I do miss some aspects in the M&M that are essential when writing according to the ARRIVE guidelines. Below, I list a few that I noticed. I advise the authors to take the ARRIVE guidelines at hand and check for any other information that may be missing:
- Full (sub)strain name of the Swiss mice, and where they were purchased or bred.
Response: The used strain of albino mice was CD1 and was obtained from the animal unit of Mansoura University, Egypt. The manuscript has been updated with this information; please see lines 114-116 of the revised version of the manuscript.
Comment: Type of diet the mice were fed
Response: The used diet was a commercial rodent pellet diet purchased from Meladco for animal feed company, El-Obour City, Cairo, Egypt. The manuscript has been updated with this information; please see line 119 of the revised version of the manuscript.
Comment: Experimental set-up: 60 mice were used, however, the experimental design only accounts for 3 groups of 15 mice each. What happened to the other 15 mice?
Response: The authors apologize for such confusion. The correct total number of animals was 60. The number of mice per group was 20, divided into four cages of five mice. The manuscript text has been corrected at relevant parts; please see lines 124-125 and 147-150 of the revised version of the manuscript.
Comment: Information on randomisation is missing. This is especially important for the randomisation of ovaries that are attributed to 3 groups, since each mouse has only 2 ovaries. It is also important for the 5 mice in each cage: were they randomly assigned to the different treatment groups, or were whole cages assigned to one group? This may affect the experimental unit. If whole cages were assigned to an experimental group, but still mice were seen as an experimental unit, some information is necessary describing how the authors interpret the risk of pseudo replication for the parameters they measure.
Response: Thank you for pointing this out. We did randomization between cages of the same group to avoid bias during sampling. However, and unfortunately, we were not able to do this between cages of different groups to avoid unintended mixing of animals from different groups during repeated injections; this point has been cleared in line 148 of the revised version of the manuscript.
Comment: Information on blinding is missing: Were people injecting the mice blind for treatment? Were people analysing the ovaries blind for treatment?
Response: All mouse injections were supervised by a member of our research team (AMA). He was also in charge of blood and tissue sampling. After this step, the collected samples were blinded by him and further processed by two other different researchers (SIR and AIA).
Comment: Information on acclimitisation is missing.
Response: Mice were allowed to acclimatize for two weeks prior to starting the experiments; please see line 124 of the revised version of the manuscript.
Comment: Statistics: for all comparisons, one-way anova with Tukeys post-hoc test were used. Did all parameters meet the assumptions for one way anova? Eg with regard to normal distribution? If not, what was done? Please provide some more info for each of the parameters measured.
Response: We would like to thank our reviewer for this comment. Yes, we have used ANOVA with Tukey for all data analysis, assumming a normal distribution pattern of data. The main reason for this was the absence of extreme outliers among the datasets of each group. We also confirmed this in the histogram section of Graphpad software prior to statistical testing.
Comment: Ethical remark: Prior to euthanasia, mice were bled through retro-orbital puncture. According to good practice, this method is only acceptable under terminal anesthesia. Was terminal anesthesia applied? If not, why not?
Response: We agree with our reviewer that terminal anesthesia was required prior to blood sampling from the retro-orbital plexus. However, we didn’t use general anesthesia in our study to eliminate possible effects of general anesthetics on serum hormone levels. Instead, we applied a potent topical eye anesthetic solution containing 0.5% tetracaine HCl to alleviate instant pain caused by the puncture. The manuscript has been updated with this information; please see lines 144-146 of the revised version of the manuscript.
Reviewer 4 Report
Comments and Suggestions for Authors
The paper show the protective effect of lycopene on the count of ovarian follicles and the production of reproductive hormones by the ovaries has been shown to be associated with enhanced ovarian antioxidant capacity and reduced rates of follicle apoptosis,and molecular biological tests also support this conclusion. If a reagent with recognized antioxidant effects can be set up for parallel experiments, it would be more valuable to obtain conclusions through comparison.
Author Response
Comment: The paper show the protective effect of lycopene on the count of ovarian follicles and the production of reproductive hormones by the ovaries has been shown to be associated with enhanced ovarian antioxidant capacity and reduced rates of follicle apoptosis and molecular biological tests also support this conclusion. If a reagent with recognized antioxidant effects can be set up for parallel experiments, it would be more valuable to obtain conclusions through comparison.
Response: We would like to thank our reviewer for the time and effort spent reviewing our work. We agree that comparing the effects gained by lycopene on mouse ovaries and serum reproductive parameters would be more interesting if the data were presented parallel to another substance with known potent effects. However, this was not planned during the present study and could be a matter of our future investigations.
Round 2
Reviewer 1 Report
Comments and Suggestions for Authors
The revised manuscript was improved.